# Granger Causality Analysis of Transient Calcium Dynamics in the Honey Bee Antennal Lobe Network

**DOI:** 10.3390/insects14060539

**Published:** 2023-06-09

**Authors:** Marco Paoli, Yuri Antonacci, Angela Albi, Luca Faes, Albrecht Haase

**Affiliations:** 1Research Center of Animal Cognition, Center for Integrative Biology, CNRS, University of Toulouse, 31400 Toulouse, France; 2Dipartimento di Ingegneria, Università di Palermo, 90128 Palermo, Italy; 3Department of Collective Behavior, Max Planck Institute of Animal Behavior, 78457 Konstanz, Germany; 4Department of Biology, University of Konstanz, 78457 Konstanz, Germany; 5Center for Mind/Brain Science (CIMeC), University of Trento, 38068 Rovereto, Italy; 6Department of Physics, University of Trento, 38123 Povo, Italy

**Keywords:** antennal lobe, Granger causality, calcium imaging, olfaction, sensory network, two-photon microscopy

## Abstract

**Simple Summary:**

In this work, we imaged the neuronal activity in the peripheral olfactory system of honey bees, the antennal lobes. Instead of the conventional analysis that focuses on the activity patterns in the network nodes, the glomeruli, we present a new approach that analyzes the causal connections between glomeruli. Our analysis shows that such links are present even in the absence of olfactory stimulation and that, upon exposure to an odor pulse, the connectivity increases and the structure of the connections becomes odorant-specific. This approach allows mapping the information flow that underlies the formation of odor-specific response maps in the antennal lobe.

**Abstract:**

Odorant processing presents multiple parallels across animal species, and insects became relevant models for the study of olfactory coding because of the tractability of the underlying neural circuits. Within the insect brain, odorants are received by olfactory sensory neurons and processed by the antennal lobe network. Such a network comprises multiple nodes, named glomeruli, that receive sensory information and are interconnected by local interneurons participating in shaping the neural representation of an odorant. The study of functional connectivity between the nodes of a sensory network in vivo is a challenging task that requires simultaneous recording from multiple nodes at high temporal resolutions. Here, we followed the calcium dynamics of antennal lobe glomeruli and applied Granger causality analysis to assess the functional connectivity among network nodes in the presence and absence of an odorous stimulus. This approach revealed the existence of causal connectivity links between antennal lobe glomeruli in the absence of olfactory stimulation, while at odor arrival, the connectivity network’s density increased and became stimulus-specific. Thus, such an analytical approach may provide a new tool for the investigation of neural network plasticity in vivo.

## 1. Introduction

Honey bees are eusocial insects that strongly rely on olfaction, e.g., for foraging, social communication, and mating. Their olfactory system has been extensively studied in the past decades, and it is structured along different anatomical and functional layers of odor processing [1]. Volatile chemicals are received by olfactory receptors located on the antennae, where they are received by ~60,000 olfactory receptor neurons (ORNs). The olfactory input is forwarded to the first processing center, the antennal lobe (AL), where ORNs bearing the same olfactory receptor converge into one of about 160 glomeruli, the functional nodes of the AL processing network. Incoming neuronal inputs are further processed by ~4000 local neurons (LNs), and the resulting signal is forwarded by ~800 projection neurons (PNs) into higher-order brain centers such as the mushroom bodies (MBs), dedicated to multisensory integration and memory storage, and the lateral horns (LHs), most likely involved in odor valence coding [1]. 

The development of in vivo calcium imaging in the honey bee [2] allowed simultaneous recording of the activity of multiple glomeruli. This allowed the description of the fundamental principles of olfactory coding, such as the stereotypical odorant-specific glomerular response maps [2], which correspond to the time-averaged PN firing rates [3], the representation of information on the odors’ chemical properties [4] and concentration [5], and the link between similarity of perception and neural representation of an olfactory cue [6]. With the implementation of fast scanning two-photon imaging, the temporal resolution has been increased by more than an order of magnitude, up to more than 100 Hz [7], thus providing access to fine temporal properties of the olfactory code. Notably, calcium sensors’ kinetics are characterized by a fast onset (<10 ms) and a slow offset (>100 ms). This provides a limitation for estimating the firing rate underneath the slow calcium transients [3,8]. Nonetheless, fast calcium imaging analysis has been proven useful in assessing differences in response latencies among neurons/glomeruli [9] and their oscillatory properties [10], and it may be informative when monitoring correlation and information transfer among network nodes. 

Interglomerular connectivity has been largely investigated by means of genetic, physiological, pharmacological, and modeling approaches. In Drosophila, genetic silencing of all ORNs converging onto one glomerulus revealed that the post-synaptic projection neurons (PNs) of such a glomerulus received lateral excitatory input from other glomeruli [11], suggesting that a network of excitatory lateral connections distributes odor-evoked excitation among glomeruli. Electrophysiological and imaging analysis revealed that excitatory local neurons receive monosynaptic input from ORNs [12] and that reciprocal dendrodendritic chemical and electrical synapses exist between LNs and PNs in the fruit fly AL [12,13]. In honey bees, there is no evidence so far of the existence of excitatory local neurons in the AL. The organization of the honey bee AL local inhibitory network was investigated by combining calcium imaging with targeted neuropharmacology [14]. Such an approach allowed for characterizing the strength and spatial pattern of the inhibitory intra-glomerular connections, showing that connectivity is patchy and does not necessarily follow a proximity rule. Other studies relied on a modeling approach aimed at matching ORNs and PNs signals to investigate the transformation mediated by the local inhibitory network [15]. This provided further information on the nature of inter-glomerular coupling by showing that glomerular pairing is based on odor tuning rather than on spatial relationships and that such an arrangement optimizes odor contrast in a honey bee’s AL.

Here, we propose an alternative approach to the study of the antennal lobe network, which is to investigate its functional connectivity using Granger causality [16]. Granger causality (GC) measures the relationship among different time series by assessing if the past of time series X can help predict the future of time series Y. It relates the presence of a cause–effect relation to two aspects: the cause must precede the effect in time and must carry unique information about the present value of the effect. This relationship is not symmetrical and can be bidirectional, thereby enabling the detection of directed and reciprocal influences [17]. Such a measure is not alternative but complementary to the widely used Pearson’s correlation, which is a measure of association between time series that does not assume causality between them, but rather seeks increasing and decreasing trends in the data, representative of the statistical dependencies between them [18]. The use of Granger causality has been recently coupled with calcium imaging analysis for assessing functional connectivity in vivo. A study on the mouse auditory cortex showed that such an approach is suitable for detecting the distance and density of inter-neuronal connections, which are in agreement with empirical observations [19]. A more recent study used a GC approach to analyze the interaction between motoneurons in the brainstem of zebrafish larvae [20], showing its applicability for the identification of driver and target neurons within a network in a living animal.

Here, we used calcium imaging analysis of AL glomeruli during olfactory stimulation to acquire the time series of multiple nodes of the AL network for different subjects and odorants. Then, we identified the network of functional glomerular coupling based on Granger causality analysis, and we measured how the functional connectivity changes across individuals, for different stimuli, and across different phases of odor exposure. Finally, applying a template recognition test, we assessed the information content of the functional connectivity network across odorants and subjects.

## 2. Materials and Methods

### 2.1. Bee Preparation

Forager honey bees *Apis mellifera* were collected at the outdoor hives of the institute (CIMeC, Rovereto, Italy) on the day of the experiment (*n* = 15). They were cold-anesthetized at −20 °C, placed on custom-made Plexiglass holders, and their heads were fixed with soft dental wax to avoid movements [21]. After opening a small window in the head cuticle, glands and tracheas were gently displaced to expose the bee’s brain, and the tip of a pulled glass capillary coated with Fura-2-dextran 10 kDa (ThermoFisher Scientific, Waltham, MA, USA) was inserted between the medial and lateral mushroom body calyces to label the projection neurons’ tracts [22]. Then, the head capsule was closed and sealed with n-eicosane (Sigma-Aldrich, St. Louis, MO, USA) to prevent brain desiccation. Bees were fed with 20 μL of a 50% sugar/water solution and kept overnight at room temperature in a dark and humid environment. On the following day, the cuticle window was re-opened, and the antennal lobe ipsilateral to the injection site was exposed for imaging. The brain was covered in transparent silicon (Kwik-Sil, WPI, Sarasota, FL, USA) to prevent brain movements. Note that this procedure has an intrinsic variability that depends on the amount of Fura-2 delivered by the experimenter and on the number of labeled projection neurons. Still, multiple studies have shown convergent results on odor-elicited glomerular responses, indicating that such a procedure remains valid to investigate olfactory processing in the honey bee antennal lobe [9,23,24,25,26].

### 2.2. Olfactory Stimulation

Olfactory stimuli were delivered with a custom-built olfactometer comprising eight odor channels and controlled via a custom-made LabVIEW (National Instruments, Austin, TX, USA) interface, which synchronized stimulation protocol with the imaging acquisition [21]. For this study, we used six ecologically relevant odorants known to elicit distinct responses in the antennal lobe glomeruli: 1-hexanol (1HEX), 3-hexanol (3HEX), 1-nonanol (1NON), isoamyl acetate (ISOA), acetophenone (ACTP), and benzaldehyde (BZDA) (all from Sigma-Aldrich) [2,9]. Additionally, they provide different degrees of structural and functional variability: 1-hexanol and 3-hexanol have the same carbon length but vary for the position of the hydroxyl group; 1-hexanol and 1-nonanol are primary alcohols with different chain lengths; acetophenone and benzaldehyde both have a benzene ring, but with a ketone and an aldehyde group, respectively; isoamyl acetate is the major component of the honey bee alarm pheromone [27], while all other odorants are typical floral scent components. Stimuli were presented in a 1 s ON/9 s OFF protocol and each odorant was delivered for 30 consecutive trials. All odorants were diluted 1:200 in mineral oil (Sigma-Aldrich). These concentrations were chosen to be well above the receptor sensitivity threshold and below the saturation level [5].

### 2.3. Calcium Imaging Acquisition and Data Processing

Optical imaging was performed via a two-photon imaging platform based on an Ultima IV microscope (Brucker, Billerica, MA, USA). The calcium sensor Fura-2 was excited at 800 nm and fluorescent changes were recorded with a photomultiplier (Hamamatsu, Iwata City, Japan) at 525 ± 20 nm. Images were acquired via line scanning along a user-drawn scan path crossing all glomeruli in the imaging field with a scan rate of 100 Hz. Individual glomeruli were identified and labeled according to the honey bee reference atlas [28] (Figure 1). To allow for cross-subject comparisons, the analysis was limited to the 10 glomeruli that could be identified, with a few exceptions, in all imaged bees. For the analysis, a whole imaging session, comprising 30 repetitions of 6 odorants, was decomposed into individual stimulation trials. The average fluorescence signal change Δ*F* in each identified glomerulus for each stimulation trial was normalized with respect to the pre-stimulus baseline *F* (averaged over 1 s). An increase in calcium decreased the Fura-2 fluorescence intensity. To still represent excitatory responses as positive changes and inhibitory responses as negative changes, data are always presented as the negative relative fluorescence change (−Δ*F*/*F*) (Figure 2A).

### 2.4. Granger Causality

The temporal statistical structure of the calcium time series recorded from ten glomeruli (*M* = 10) for each animal was investigated using a vector autoregressive model (VAR) representation. Specifically, at each time instant *n*, denoted as the present state, the present state of the process modeling the time series from all glomeruli, $\;Y_{n}={\left[Y_{1,n},\ldots ,Y_{M,n}\right]}^{T}$, was described as follows:(1)$$Y_{n}=\overset{p}{\underset{k=1}{\sum }}A_{k}Y_{n-k}+U_{n},$$
where $A_{k}$ is an M×M matrix containing the coefficients that relate $\;Y_{n}$ to the past state of the process evaluated at lag k, $Y_{n-k}$, *p* represents the model order, and $U_{n}={\left[U_{1,n},\ldots ,U_{M,n}\right]}^{T}$ is a vector of M zero-mean white and uncorrelated input noise processes (prediction errors). Conceptually, causality relations are found when the pathway relevant to the interaction is active, that is, described by non-zero coefficients stored in $A=\left[A_{1},\ldots ,A_{p}\right].\;$ Starting from the time series of each given subject, the VAR model (1) was identified via a least squares method, finding estimates of the coefficients $A_{k}$ and of the variance of the input noises, i.e., $\lambda_{U_{m}}=\mathrm{var}\left(U_{m}\right),\;m=1,\ldots ,M$ [18]. Here, to account for indirect paths that could lead to spurious links, GC in its conditional form was computed using an approach based on state-space modeling [29]. The conditional GC measure quantifies the extent to which the past states of the driver $Y_{i}$ help in predicting the present state of the target $Y_{j}$ above and beyond its own past states and the past states of all other processes collected in the vector $Y_{k}$. The measure was computed by varying i and j in the range $\left\{1,\ldots ,M\right\}\;\left(i\neq j\right)$ to quantify the causal coupling along the two directions of interaction for each pair of glomeruli in each individual. Furthermore, to establish the existence of a link from the $i^{th}$ to the $j^{th}$ node of the observed network, the statistical significance of the computed conditional GC was tested using an asymptotic statistics approach. This approach makes use of a Fisher *F*-test applied to the estimated prediction error variances of the full and restricted models, considering the computation of multiple GCs via false discovery rate (FDR) correction (α = 0.05) [30]. After the *F*-test, each GC link became associated with a *p*-value. Non-significant estimates were then set equal to zero, while the significant ones were maintained. This resulted in a matrix of weighted GC connectivity values.

### 2.5. Obtaining GC Maps

For the analysis of the signal dynamics across time, as a first step, the time series were sub-divided into seven 1 s windows: ON phase (0–1 s), earlyOFF phase (1–2 s), and OFF phases (2–3 s, 3–4 s, 4–5 s, 5–6 s, and 6–7 s). For each glomerulus, all repetitions in the individual time windows were concatenated, and the resulting time series was used to calculate the GC relationships for each individual (*n* = 15 bees). The concatenation of 30 repetitions of 1 s each at 100 Hz sampling rate provided sequences of 3000 time points that allowed for a more reliable GC analysis. A GC analysis was performed for each bee, time window, and odorant. The obtained matrices contained statistically significant GC correlation measures between glomeruli, or zero if results were non-significant. 

Because the same glomeruli could be identified across individuals, edge-centered maps were averaged across ALs to detect GC links that were conserved between pairs of glomeruli. Note that with edge-centered maps we refer to the response maps based on the connectivity among pairs of glomeruli, in opposition to node-centered maps, which are based on the response intensity of the individual glomeruli. The number of connections was normalized from 0 (when no individual showed that connection) to 1 (all animals presented that specific inter-glomerular connection). 

### 2.6. Network Analysis

To identify the contribution of different frequency regimes to the GC connections, time series were decomposed into slow and fast components by using an infinite impulse response (IIR) autoregressive filter with zero phase [31] applied to the 30 trials. The filter was implemented first in a low-pass configuration (cut-off frequency ~1.5 Hz) to extract the slow signal components mostly related to the stimulus response, and then in a high-pass configuration (cut-off frequency ~2.5 Hz) to extract the fast components. In the latter case, before high-pass filtering, the mean response obtained by averaging the signal across the 30 repetitions was subtracted from each trial. GC analyses were then performed on both frequency components separately. 

The density of each estimated weighted network was evaluated as the fraction of significant connections to the total number of possible connections [32]. Significant changes in network density across time bins were assessed with a Kruskal–Wallis test performed separately for the unfiltered signal and the fast and slow components.

### 2.7. Stereotypy Analysis

The similarity between the edge-centered odor representations and the GC connectivity maps, within and across individuals, was quantified via a best-match-to-template analysis. For the within-subject analysis, we split the 30 olfactory stimulations into 6 groups of 5 trials. Hence, GC measures were calculated independently for each group. This resulted in 6 connectivity maps for each odorant-bee combination. Then, the connectivity map for each stimulus trial (test) was tested against the mean response map across all remaining trials for all odorants (templates). The similarity between maps of two odors *o1* and *o2* was quantified by performing a Pearson’s correlation analysis between the vectorized GC maps of odors *o1* and *o2*, after statistical validation. The best-fitting template was considered the one with the highest correlation coefficient *r*. The probability that the best fit occurred with the correct odor template—i.e., the fraction of times where a response map was best fitted to the correct odor template—is plotted in Figure 3A. 

For the across-subject analysis, the average connectivity in an odorant-bee combination (calculated across all 30 olfactory stimulations) was tested against the average connectivity maps across all individuals, excluding the tested one. For each test, the best template odorant would receive a score of 1 and all other odorants a score of 0. If the highest correlation was shared by more than one odor template, the score was split accordingly. Results in each bee and the averages across bees are shown in Figure 3C.

The same best-match-to-template test was performed for the node-centered maps, that is, the time-averaged glomerular response amplitudes, within (Figure 3B) and across (Figure 3D) individuals. 

For each test, significant differences from a random distribution, which would give 1/6 = 0.167 fit-to-template probability for each odor, were analyzed with a Wilcoxon sum rank test with FDR correction. Original and adjusted *p*-values for all tests and the signed rank values *W* are shown in Appendix A.

## 3. Results

### 3.1. Node-Centered Odor Response Maps

The spatio-temporal activity patterns for six odorants and 15 bees were first analyzed by confronting glomerular responses. This is the classical description of a stimulus representation in the AL, a node-centered network analysis. Figure 2A shows the trial-averaged response maps of the identified glomeruli across all bees, revealing their stereotyped response patterns. On average, two to four of the monitored glomeruli showed a clear excitatory response during odor arrival. This analytical approach for studying olfactory coding takes into account the responses of individual functional units, measuring physiological parameters such as glomerular response intensity and latency [4,33]. 

### 3.2. Edge-Centered Odor Response Maps

#### 3.2.1. Difference between Resting-State and Odor-Induced Connectivity

To investigate the functional connectivity among AL glomeruli, we analyzed to what extent the signal of one glomerulus influences all other glomeruli of the same network. This analysis was performed via Granger causality (GC) in its conditional form, which quantifies the amount of information contained in the present state of one glomerulus (target process) that can be predicted by the past state of another glomerulus (source process), above and beyond the information that is predicted already by the past states of the target process itself and of all others glomeruli [34]. In other words, we assessed if the activity of one node could predict (or be predicted by) the activity of all other nodes. To assess the stability of inter-glomerular connections, we calculated causal connectivity maps during and after olfactory stimulation and with six different odorants. Importantly, with connectivity or edge-centered maps, we refer to the calculated GC correlations after statistical validation (see Section 2). As shown in Figure 2B, functional connectivity emerges both in the presence and absence of an olfactory input. Nevertheless, the network density decreases significantly after odor offset, suggesting that olfactory stimulation modulates inter-glomerular causal connectivity (Figure 2B,C). 

#### 3.2.2. GC Maps Do Not Primarily Reflect Glomerular Response Similarity 

A comparison between node- and edge-centered maps (Figure 2A,B) revealed that causal connections are more likely to link strongly responsive glomeruli. To test whether GC depends on glomerular response patterns similarity, rather than on information transfer across nodes, we used two complementary approaches. First, we filtered glomerular activity traces to have time series containing only slow and fast signal components (see Section 2): the former component is dominated by slow stimulus-induced transients, while the latter comprises only fast oscillations (Appendix A). If the network states were dominated by the similarity between glomerular responses, the links between highly responsive glomeruli should be present in the slow signal, deprived of fast oscillatory components. However, GC analysis of the slow components provided a highly reduced number of edges, advocating against an equivalence of node- and edge-centered odor representations (Figure 2C). Conversely, when considering only the fast components of a time series, the network density remains comparable to the one obtained by analyzing the unfiltered signal. This indicates that, in our case, GC connectivity relies mainly on fast signal components. The analysis of connectivity map dynamics (Figure 2C) showed that the average network density remains high also after odor offset, although revealing a significant decrease 1 s after stimulus termination (Kruskal–Wallis test, unfiltered signal, trial effect *p* = 0.0021, *χ*^2^ = 20.8, *df* = 6). Interestingly, such an effect can be observed mainly in the slow signal components (*p* = 8.9 × 10–13, *χ*^2^ = 68.4, *df* = 6), while no change could be detected in the fast component network density (*p* = 0.89, *χ*^2^ = 2.38, *df* = 6).

To control for the possibility that the identified connections were dependent on response similarity, we generated artificial AL networks by shuffling the glomeruli across AL (Appendix A). Because of the stereotypy of glomerular responses across individuals, the odor response maps of the ‘scrambled’ ALs remain stereotyped, but the individual glomeruli are extracted from different biological entities. Performing the GC analysis on such a dataset yielded connectivity maps with a dramatic decrease in network density across bees and stimuli (Figure 2D), indicating that similarity between glomerular response profiles is not sufficient to produce the observed causal functional connectivity. Notably, a decrease in connection density upon glomerular shuffling does not indicate that connectivity maps are not conserved across animals, but that GC does not identify significant links among nodes that do not belong to the same biological network.

In conclusion, this analysis indicates that GC connectivity across AL glomeruli is not driven by response similarity, but it is often established among strongly responsive glomeruli. Moreover, it suggests that the observed causality links are detected only in real biological entities and are dominated by fast signal components.

### 3.3. Causal Functional Connectivity Contains Odor Information

Because GC connection density is higher during odorant arrival (ON phase) than after its termination, we investigated if the edge-centered maps provide stimulus-related information. Moreover, the difference among the average edge-centered maps during the OFF phase (Figure 2B, bottom row) may suggest that stimulus-related information persists several seconds after odor offset. For this, we assessed if the connectivity maps elicited by one odorant could correctly predict odor identity within and across individuals and at different time windows (see Section 2).

First, because each individual received 30 stimulations with each odorant, we split each glomerular time series into six series, each comprising five olfactory stimulations with the same odorant. Then, we performed the GC analysis on each group of trials, obtaining six independent connectivity maps for each bee/odorant combination. This procedure allowed for assessing if odorant-induced connectivity maps are conserved at the individual level. These tests were performed on the connectivity matrices calculated for three time-windows: during olfactory stimulation (ON, 0–1 s window), immediately after (earlyOFF, 1–2 s), and 5 s after odor offset (OFF, 6–7 s) (Figure 3A). Template recognition tests yielded a correct odor identification rate well above the chance level during the ON phase. However, the rate of correct identifications was reduced after odor offset-albeit significant for four odorants out of six-and decayed to chance level in the OFF window. For comparison, we performed the same analysis on the node-centered glomerular response patterns (Figure 3B). As expected [4,9,26], stimulus responses within individuals are highly reproducible, resulting in a correct template recognition in the majority of the cases. 

Next, we investigated if edge-centered maps were preserved across individuals, i.e., if stimulus-related information exchange in the functional connectivity maps was similar at the population level. In this case, a single connectivity map was calculated for each bee/odorant combination for assessing if such a map could successfully identify the correct subject-averaged odor template. Notably, the template recognition test across bees showed higher variability than that within animals. Nonetheless, it was still significantly different from random matching for three stimuli (1-hexanol, 1-nonanol, and isoamyl acetate; Figure 3C). Again, we compared this result to the same test performed with the node-centered maps. Figure 1A suggests that the latter should be rather well-preserved across animals. Still, a correct template recognition test across individuals revealed that only three out of six odorants provided a response map that allowed identifying the correct template above chance level (3-hexanol, 1-nonanol, and isoamyl acetate; Figure 3D).

## 4. Discussion

Understanding how information flows between functional units of a neural network in vivo represents a challenge in neuroscience. Granger causality has been proposed as an effective measure for identifying dynamic interactions. This study shows that GC can be applied to AL calcium imaging analyses to acquire information on the functional connectivity of the network, as well as its stimulus- and individual-specificity. This approach showed that functional connectivity among glomeruli increased during olfactory stimulation and that such connectivity occurred essentially only among nodes belonging to the same biological entity. Moreover, although network edges tend to arise among strongly responsive glomeruli, this analysis showed that response similarity was not sufficient to determine a causal correlation between the network nodes. This confirms previous reports that glomeruli displaying a strong odorant-induced response tend to fire more synchronously, possibly increasing their reciprocal excitatory connections [35]. 

Depending on the nature of the odorant, GC analysis resulted in a link density of about 12%. This means that ~88% of possible interglomerular connections were silent. A previous report in *Drosophila* [36] revealed that while PNs innervating the same glomerulus can be highly synchronized during olfactory stimulation (an effect largely due to the simultaneous arrival of the input from hundreds of ORNs activated by the same odorant), neurons innervating different glomeruli were found to be only weakly correlated. Nonetheless, such a conclusion was based on only five recordings and focused on co-occurring neuronal spikes, neglecting lateral inhibition effects. 

Antennal lobe projection neurons fire following an oscillatory pattern, which is dependent on the inhibitory activity of the local interneurons [37]. Computer simulation of a PN–LN network showed that lateral inhibition performed by LNs could entrain PNs to spike in synchrony [38,39], and that subsets of coupled LNs can promote the formation of PN clusters going in and out of sync based on the nature of the olfactory input. The spatial connectivity of the local inhibitory network and the strength of the LN–PN coupling influenced the degree of the PN synchronization [38,39]. Interestingly, picrotoxin down-regulated odor discrimination ability, affecting spike synchronization rather than the overall spike discharge (i.e., the PN odor response maps). This suggests that LN-mediated information transfer across glomeruli may provide a channel for olfactory coding [40,41]. The use of pharmacology to study the influence of the inhibitory local network on GC connectivity will certainly provide us with further information on the mechanisms leading to inter-glomerular information transfer. 

To investigate the logic of connectivity among glomeruli, Linster et al. combined response amplitudes of AL input and output neurons and simulated different glomerular coupling scenarios. In agreement with our empirical observations, their modeling analysis showed that experimental data could be replicated with a functional model in which coupling was weighted by the similarity of the glomerular response profiles [15]. Additionally, the observed GC connectivity was dominated by functional links among similarly responsive glomeruli. To verify if response similarity without coupling could produce the same results, we generated artificial antennal lobes, which had a similar glomerular response profile, but whose glomeruli were taken from different individuals. Computing GC links on such artificial networks produced a dramatically low number of connections. This suggested that response profile similarity is not sufficient to determine a causal functional link, a feature that can be observed only among nodes belonging to the same biological network. Previous studies highlighted the presence of a 20 Hz oscillatory cycle in the locust’s MB, which arises from PN activity synchronization and can support olfactory coding [37]. Moreover, signal analysis of honey bee AL PNs revealed modulation by a 3 Hz oscillatory pattern during odor arrival [10]. Here, we observed that fast signal components provide for the majority of the identified connections and are sensitive to the glomerular shuffling procedure (Figure 2C,D). These observations coherently suggest a role of fast neural oscillations in olfactory coding.

From a technical standpoint, it is worth noticing that different bees may display different animal-specific response latencies in their glomerular activities. Thus, assembling artificial AL networks with glomeruli from different bees may result in a consistent (positive or negative) time shift in the response of some glomeruli. While this could negatively affect a correlation analysis, it would not impair GC analysis, which measures the influence of the past on future activity. On the contrary, a systematic time lag between the activity of two nodes would bias the system towards the creation of connectivity artifacts. Therefore, the decrease in connection density upon glomerular shuffling argues even more strongly for the physical nature of such functional links and for the fact that glomerular response latency—in scrambled AL networks as well as in real ones—is not sufficient for establishing an edge between nodes. Moreover, in our preparations, motion artifacts due to muscular contraction and hemolymph movements were limited by stabilizing the bee head and body and by enwrapping the brain with a transparent silicon [21]. Nonetheless, they may occur and provoke a general displacement of the field of view and a change in luminosity across all recorded glomeruli. However, the hypothesis that causality links could be due to motion artifacts should be rejected on the basis that this would not increase single links, but would create an offset to all connections in the matrix due to the coherence of the motion/luminosity changes across all glomeruli.

Granger causality connectivity maps showed that networks are not random but provide stimulus-specific information. The odorant specificity of GC networks was assessed at the individual level—i.e., how odor-specific they are within an individual AL across single trials—by testing if the connectivity profiles calculated for the same bee but for different olfactory stimulations were conserved enough to allow identifying the correct odorant. Correct identifications were well above chance level, thus advocating that information transfer among glomeruli carries stimulus-specific information. Similarly, we assessed causality map stereotypy across individuals. Despite the stereotypical responses of glomeruli across individuals, quantitative match-to-template tests did not allow for a general error-free odorant prediction for both node- and edge-centered maps. This was likely limited by the small sub-population of AL glomeruli we were able to identify with certainty. In fact, in a larger subset of glomeruli, each odorant would likely have some strongly responsive units in its response map, possibly leading to a better match-to-template probability of the node-centered representation maps. Similarly, a larger subset of glomeruli could provide more stimulus-specific functional connections, resulting in more odorant-specific edge-centered maps. On the other hand, it is also possible that the local connectivity is different among individuals, thus resulting in non-stereotyped connectivity maps at the population level. In fact, the inter-individual variability of stimulus-specific connectivity maps is coherent with physiological findings showing that local AL connectivity is not pre-determined but is susceptible to a certain variability, which may depend on the difference in developmental plasticity and/or past experience [14,42]. Predictive power was strong during the ON phase, but quickly decreased after odor offset to disappear completely in the OFF window. The residual predictive value observable across GC maps during the first second after odor offset (i.e., earlyOFF) was possibly due to stimulus-specific glomerular responses lasting longer than the stimulation itself, or to neurons that were quickly activated at odor offset. These responses generally lasted <2 s (see Appendix A), thus explaining the disappearance of any odorant specificity in the OFF phase. 

Our experimental design included odorants of different natures to investigate general features of functional connectivity during olfactory coding. Among them, the alarm pheromone isoamyl acetate has an intrinsic valence and elicits stereotyped behaviors across bees [43]. Thus, a more conserved functional connectivity map could have been hypothesized. However, the odor predictability of isoamyl acetate is comparable to the ones of hexanol and nonanol, and greater than those of benzaldehyde and acetophenone (Figure 3). Although not conclusive, this may indicate that stimulus information content in the AL network may not depend on stimulus valence. On the contrary, it may be similar across stimuli and the different level of odorant predictability may depend on the pool of glomeruli included in the analysis.

Notably, the concept of connectivity encompasses various modalities of interaction between different anatomical regions, which could be anatomical or functional. In the former case, connectivity was intended as a representation of anatomical fiber pathways and can be intended as a purely physical phenomenon. In the latter case, the functional connectivity was defined in terms of the statistical connections between the dynamic activity of neural units in different anatomical locations and, according to the original definition, does not relate to any specific direction or structure of the analyzed network. Instead, it is purely based on the probabilities of the observed neural responses [44]. With these premises, the characterization of the AL functional connectivity network we provided should not be interpreted as direct synaptic connections between glomeruli. Our analysis revealed a number of sparse interglomerular connections that varied dynamically according to the odorant used and the phase of the stimulation. While it is possible that in some cases GC links may be supported by anatomical connections between glomeruli, the flexibility of the measured functional connectivity suggests that it may rely on other mechanisms, such as synchronization and desynchronization of glomerular oscillatory activity. Finally, it is important to bear in mind that the appearance and vanishing of interglomerular connections cannot be readily interpreted as an activation or a blockage of information transfer between them. Such fluctuations can be strongly influenced by the FDR correction procedure adopted to assess for statistical significance of the detected GC connections. Thus, the disappearance of a link may simply reflect the fact that the available data cannot support the existence of such a link with the desired level of statistical confidence.

All in all, this work has shown the feasibility of using Granger causality for exploring functional connectivity across nodes of the AL network in vivo. It confirmed some of the connectivity rules previously observed with modeling approaches [14,15,35,38] and suggested that such connectivity is dynamic and may carry odorant-specific information. In the future, the use of pharmacology will be crucial in understanding the neurochemical nature that allows information transfer across glomeruli, and in particular the role of the local GABAergic interneurons [45].

## Figures and Tables

**Figure 1 insects-14-00539-f001:**
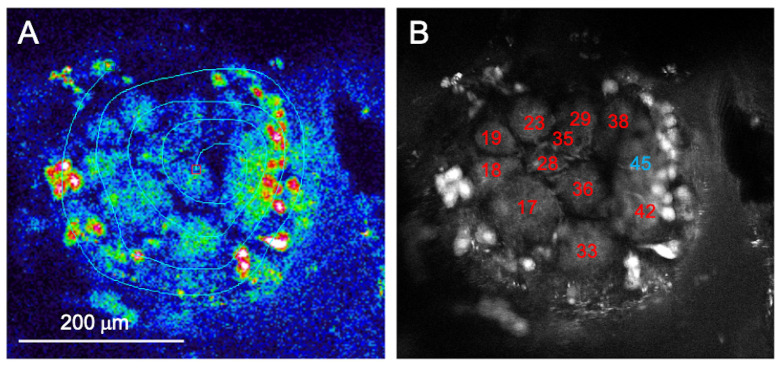
Identification of antennal lobe glomeruli. (**A**) A fixed plane of the antennal lobe was iteratively scanned along a custom-drawn line crossing multiple glomeruli. (**B**) After functional data acquisition, a higher resolution *z*-stack of the antennal lobe was acquired and used for glomerular identification. Labeling was based on glomerular size, shape, and location according to the atlas of honey bee ALs [28]. In red, identified glomeruli belonging to the T1 antennal lobe tract. Glomerulus 45, in blue, is innervated by the T3 tract.

**Figure 2 insects-14-00539-f002:**
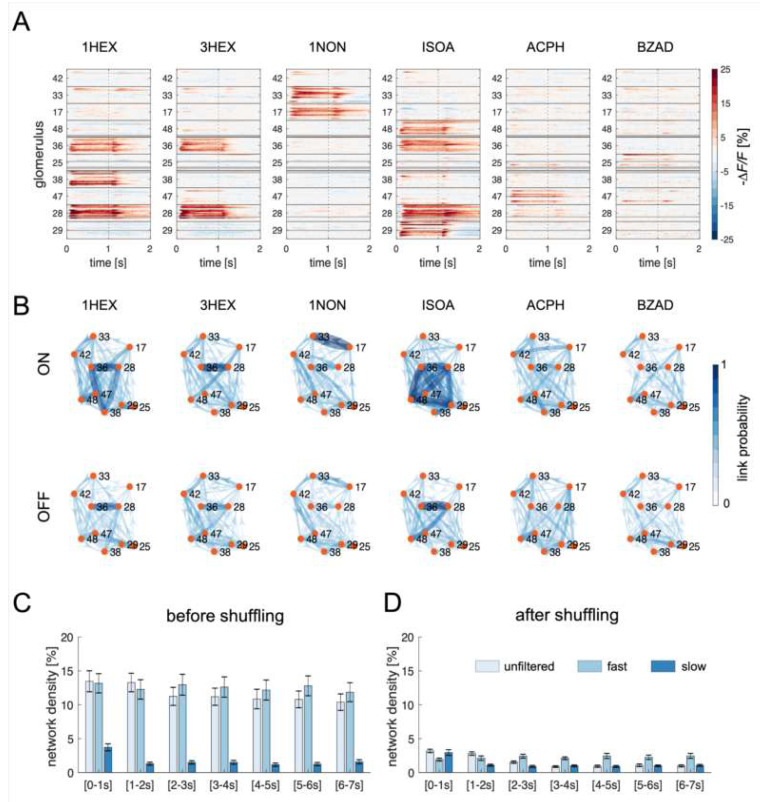
Node- and edge-centered odorant response maps. (**A**) Glomerular responses across bees and glomeruli. The relative fluorescence change across time is color-coded; gray lines represent the unavailability of individual glomerular data in single bees. Olfactory stimulation was delivered in the 0 to 1 s interval. The *y*-axis shows the response profiles of individual bees (*n* = 15) grouped according to the glomerulus ID number. (**B**) Mean connectivity maps across all bees calculated during stimulation (*t* = 0 to 1 s, ON, top row) and 5 s after odor offset (*t* = 6 to 7 s, OFF, bottom row). Link thickness and color darkness indicate the probability of link detection across all analyzed bees. Glomeruli are shown as orange nodes and are identified by ID number (**C**) Mean (±S.E.M.) network density averaged over all bees, odorants, and time for connectivity maps computed from the unfiltered signal, the fast, and the slow signal components. (**D**) Network density of the connectivity maps calculated from the same dataset after shuffling glomeruli across individuals. Abbreviations: 1-hexanol, 1HEX; 3-hexanol, 3HEX; 1-nonanol, 1NON; isoamyl acetate, ISOA; acetophenone, ACPH; benzaldehyde, BZAD.

**Figure 3 insects-14-00539-f003:**
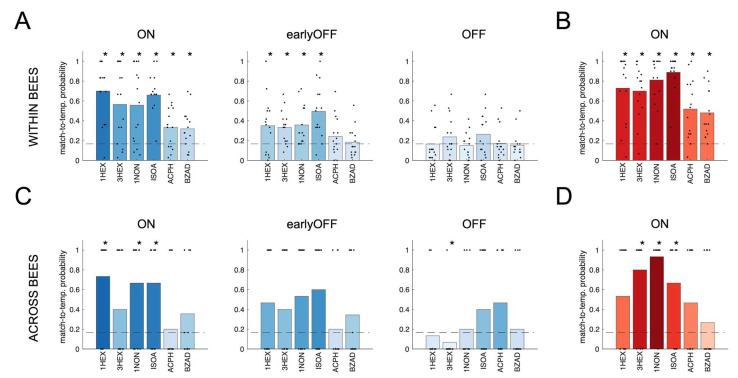
Best-match-to-template tests for edge-centered connectivity maps (**A**,**C**) and for node-centered glomerular response maps (**B**,**D**). (**A**) Within each bee, 6 connectivity maps were calculated for each odorant, each based on 5 stimulus repetitions. Each connectivity map (test) was tested against the mean map for each odorant (templates), except the tested one. (**B**) Within each bee, the glomerular response map for each stimulus trial (test) was tested against the mean response map across all trials, except the tested one for each odorant (templates). (**C**) Across bees, the trial-averaged connectivity profile for a single odorant/bee combination (test) was tested against the mean connectivity profile of the individual odorants averaged across all bees except the tested one (templates). (**D**) Across bees, the trial-averaged glomerular response map for each odor–bee combination (test) was tested against the mean response maps of each odorant averaged across all bees except for the tested one (templates). In all subplots, dots indicate the test-to-template matching probability for each bee (*n* = 15); bars indicate the average value across bees. The horizontal dashed line represents the chance level. Significant differences from a random distribution were determined via a Wilcoxon signed-rank test with FDR correction (* indicates adjusted *p*-values < 0.05, exact values in Appendix A).

## Data Availability

All data will be made available upon request to the corresponding authors.

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
