# Peer review of "Granger Causality Analysis of Transient Calcium Dynamics in the Honey Bee Antennal Lobe Network"

_insects, 2023, doi:10.3390/insects14060539_

Round 1

Reviewer 1 Report

In this manuscript, the authors investigated network connections among glomeruli in the honey bee antennal lobe by using calcium imaging and GC analyses. The antennal lobe of insects has been a good model system for studying odor representation in the brain. The majority of the former studies have focused on the activity patterns of the glomeruli, which is referred to as conventional analysis in the text, and so far relatively less effort has been made for investigating the response dynamics. In this study, the authors present a new approach for causal analyses of the odor responses of the multiple glomeruli, which greatly expands our possibility for understanding the olfactory coding system in animals. The study is quite original and would attract the interests of readers not only in the field of insect neuroscience but also in other wide fields of research. I do not have any concerns about the research methods and data itself presented in the manuscript but I would like to list some points below, which I recommend to add descriptions in the text.

1) The results indicate that the causal connectivity between the glomeruli relies mainly on the fast oscillatory components in the odor response. As far as I understood, the odor-specific oscillatory LFP responses in the insect brain, which has been reported by Laurent’s group, could be detected in the mushroom body (MB) calyces. I think the present analyses for the fast/slow components are based on this knowledge. As the authors mentioned in the text, Laurent’s group has reported that the LFP oscillation depends on synchronous activity patterns in the projection neurons (PNs), but it also reflects the activity of the MB intrinsic neurons. In the present study, the calcium responses in the PNs recorded in the antennal lobe (AL) and I think it is not comparable to the LFP in the MB. In addition, the oscillatory response in the AL has been reported, too. Please summarize the essence of these studies and explain their relationship with the present study.

2) The connectivity maps calculated 5s after the stimulus (Fig. 1B below) imply the network is also odor-specific even after the stimulation but it is not mentioned in the text at all. Please give some explanations about these odor-specific patterns during the OFF status. I think some projection neurons remain activated for a while after the offset of the stimulus. Does it affect the current results?

3) As described in Materials and Methods, the odorants for the experiments were chosen as providing structural and functional variability, although it does not mentioned in the Results and Discussion. In my opinion, it would be better to add some discussions from this perspective. In Fig. 2C, the responses to 1-hexanol, 1-nonanol, and isoamyl acetate are relatively similar across individuals. Is there any functional difference between these three odors and others? For example, for me, it seems to be reasonable that the pheromonal component elicits similar responses among individuals because it can cause stereotypic behavior.

Minor points

1) The first three lines in the result section should be removed.

2) I found some words that should be in italics. Please check it again throughout the text.

Reviewer 2 Report

The paper entitled " Granger causality analysis of transient calcium dynamics in the honey bee antennal lobe network” analyzed neuronal activity in the antennal lobes and presents a new approach that analyses the causal connections between glomeruli. This approach allows mapping the information flow that underlies the formation of odor-specific response maps in the antennal lobe.

The analytical method used in this study is relatively novel and innovative, which provides new ideas for a more macroscopic analysis of the olfactory neural coding of AL. I have a few questions that need to be clarified by the author:

1.     The correlation map between different glomeruli reactions was analyzed, so how did the experiment determine that the regions imaged between different bees contained the same glomeruli? Are there any anatomy results that prove recorded areas of the different bees are from the same glomeruli?

2.     The results showed the existence of causal connectivity links between AL glomeruli even in the absence of olfactory stimulation. And the conclusion is illustrated in Figure 1B, connectivity maps for the odor offset. To get this conclusion as the author said, the response should theoretically not be related to the corresponding odor stimulus in the off-stage of each odor stimulus. However, from your Fig1B-OFF results, it seems that the responses of the network still have a certain correlation with the corresponding odor stimulus, such as OSOA odor, in the On-stage, the connection between 36 and 28 is relatively strong. And in the corresponding OFF stage, the connection between 36 and 28 is also stronger than the other connections. This may be related to the ON stage response of odor stimulus, such as the adaptation of the response to each odor stimulation. So, is there a direct recording of spontaneous activities from AL glomeruli that does not involve olfactory stimulation? It can more intuitively reflect the existence of causal connectivity links between AL glomeruli.

English language is good enough for the publication

Reviewer 3 Report

This is a well written paper studying neuronal activity in the honey bee antennal lobe network using Granger causality analysis of time series data on node activity collected from imaging experiments. However, the methodology is quite involved and some aspects of the paper could be clearer.

Some points for correction and clarification are listed below.

L109: please italicise Apis mellifera.  Also, what size of sample was collected?

L164 and following: define subscript p.

L165: delete comma after “relate”.

L195: explain what you mean by an edge-centered map.

L210: explain more the estimated weighted network.

L224: what do you mean by the probability of the best match?

L418: change “opposite” to “contrary”.

L240-242: the authors have forgotten to take out this note.

L485: this refers to a video but it is unclear where this is.

References: The proceedings name in reference 31 needs some abbreviation.

Figure S3, line 23: change “maps” to “map”.

Additionally, the structuring of the Results section could be clearer, as the same figures are referred to repeatedly across different sections. It would be easier to follow if one figure was discussed, then another.

Round 2

Reviewer 3 Report

The authors have attended to all of my comments, and also added extra explanation. This has improved the paper.